# Meals in Shelters during Noto Peninsula Earthquakes Are Deficient in Energy and Protein for Older Adults Vulnerable to the Disaster: Challenges and Responses

**DOI:** 10.3390/nu16121904

**Published:** 2024-06-17

**Authors:** Takamitsu Sakamoto, Kyoka Asano, Hiroyo Miyata, Teruyoshi Amagai

**Affiliations:** 1Department of General Medicine, Fukuoka Tokushukai General Hospital, Fukuoka 816-0864, Japan; 2The Graduate School of Medical Sciences, Kumamoto University, Kumamoto 860-8556, Japan; 3Tokunoshima Tokushukai General Hospital, Kagoshima 891-7101, Japan; sekaino.asano@icloud.com; 4Department of Clinical Nutrition, Kindai University Hospital, Osaka 589-8511, Japan; 5Department of Clinical Engineering, Faculty of Health Care Sciences, Jikei University of Health Care Sciences, Osaka 532-0003, Japan; amagait@yahoo.co.jp

**Keywords:** disaster, TMAT, aging society, welfare shelter, disaster nutrition

## Abstract

Background: Japan is a country often subject to natural disasters, influenced by a rapidly increasing aging demographic. The current research aims to analyze the food distribution for elderly evacuees who were relocated to a care facility in Wajima City, administered by the non-profit organization Tokushukai Medical Assistant Team (TMAT), post the Noto Peninsula Earthquake on 1 January 2024. A significant portion of the shelter’s inhabitants were elderly individuals. Methods: TMAT’s operations began immediately after the calamity, concentrating on evaluating the nutritional content of meals provided during the initial and subsequent phases, along with a thorough nutritional assessment. During this process, researchers examined the meal conditions for evacuees, including the elderly and those with disabilities, observed the actual meal distribution at welfare centers, and discussed the challenges and potential solutions. Result: Throughout the TMAT mission, a total of 700 evacuees received assistance, with 65% being 65 years old or above. An analysis of the nutritional content of the 10 meal varieties served at the shelter revealed inadequate energy and protein levels for elderly individuals, particularly men, indicating the need for future enhancements. Conclusions: Following a detailed evaluation of TMAT’s response to the Noto Peninsula earthquake, it was determined that the food provided in the shelters in the affected areas did not meet the nutritional needs of elderly individuals, especially men, based on nutritional analysis. To stress the importance of establishing an effective framework, it is recommended to promptly revise the emergency food provisions for the elderly population, considering they constitute the majority of the affected individuals.

## 1. Introduction

On 1 January 2024, an earthquake occurred at 4:10 PM on the Noto Peninsula. The non-profit organization (NPO) Team Medical Assistance Team (TMAT) promptly initiated the process of collecting relevant data (Figure 1).

Subsequently, following the completion of the registration of the medical team, they proceeded to the Disaster Medical Assistance Team (DMAT) activity center located in Nanao City, located 60 km south of the epicenter in Wajima City, to gather additional information. Later, the team traveled to Wajima City and reached the city hall at approximately 4:00 PM the next day. Following consultations with the city hall personnel and local medical practitioners, TMAT set up a makeshift clinic in an evacuation center and began executing the subsequent tasks.

1: Providing medical care at the temporary clinic;

2: Conducting rounds to provide medical care to evacuees in the shelter (including deep venous thrombosis screening);

3: Conducting rounds to provide medical care at other nearby evacuation shelters;

4: Improving the environment within the evacuation shelter (zoning, toilet cleaning, etc.).

In this study, we share the real data collected on the day of this disaster, show the results, and propose the issues to be resolved to improve the meals provided in the shelters, especially for older adults, as the majority of victims.

## 2. Subjects and Methods

The dispersal of elderly and disabled evacuees (requiring care) across various floors of the shelter led to the designation of the first floor of the facility as a “welfare evacuation center” aimed at offering round-the-clock assistance for high-risk groups, such as the elderly and individuals in need of special care. Considering the emergence of infectious diseases within the shelter, isolation units were set up, along with the implementation of infection control protocols. Medical attention was delivered at the temporary clinic, catering to around 700 evacuees within the shelter. Furthermore, to facilitate the functioning of the welfare evacuation center, the assessment of meals distributed and volunteers’ efforts were carried out to provide nutritional and dietary aid to disaster victims, hence mitigating oral vulnerability among the impacted populace.

The term “high-risk group” is used in the Cabinet Office’s Disaster Response Evacuation Shelter Operation Guidelines to refer to individuals with high levels of vulnerability, also described as “frail” individuals. These groups include the elderly, people with disabilities, pregnant women, infants, and those with chronic illnesses [3].

A methodology based on food photography was employed to assess the quality of food provided in shelter facilities [4]. Accuracy was ensured by double counting the victims’ number by a physician and a nurse. Moreover, to ensure the accuracy of nutritional analysis, the analyzed results were double-checked by two independent dietitians.

## 3. Results

Following the disaster, more than 700 individuals who had to evacuate sought sanctuary in the designated shelter, where TMAT promptly commenced the provision of medical assistance. Subsequently, around 700 patients received medical attention, with 56% female (Table 1) and 65% in the age group of 65 years or above (Table 2). The municipality of Wajima City had already been grappling with a swift escalation in its populace [5], as evidenced by an elderly population ratio of 46.2% in 2020 (in contrast to the national mean of 28.6%). These data suggest that the percentage of elderly individuals within the shelter was considerably higher.

The proportion of evacuees aged 65 and over at the welfare evacuation center was 65%, significantly higher than the national average of 28.6% and even higher than the older adult population rate of 46.2% within Wajima City itself.

The welfare evacuation center accommodated more than 20 evacuees initially, and the distribution of meals commenced a few days post the disaster. Nevertheless, the rations primarily comprised cup noodles, bread, and canned items. It was not until roughly a fortnight later that consistent meal provision within the shelter was established, instigated by the volunteering culinary endeavors.

As a result of the incomplete restoration of water and sewage infrastructures, along with the substantial older adult population, even within the welfare evacuation center, the dispensed meals were frequently served cold, with curry rice emerging as a popular choice. An assessment of the nutritional value of the primary distributed meals was carried out (see Table 3).

The nutritional values of 10 different types of food that are being distributed in evacuation centers are listed in this table.

Table 2, mentioned in the previous text, refers to meal plans designed for healthy individuals. However, the actual meal distribution during the disaster was often irregular, with inconsistent portion sizes.

Assuming that three meals with the nutritional content outlined in the tables were provided daily to evacuees throughout the disaster, the average daily intake would have been approximately 1295 kcal of energy and 47.8 g of protein. This calculation is based on an average of 120 different meal combinations from the 10 different dishes served three times a day over 10 days.

When juxtaposed with the Ministry of Health, Labor and Welfare report regarding stockpiled emergency sustenance [5], these values demonstrated that the mean energy and protein content available in social assistance establishments stood at 1200 kcal/day and 36.8 g/day, respectively. In comparison to the nationwide study, it was observed that the energy content in social assistance establishments represented 8% of the total energy, 30% of the protein, and 30% of the national survey [6]. This outcome surpasses the mean value recorded in the national survey.

In this situation, even assuming that they ate all three meals a day of the food provided, their energy consumption remained at 61% and 78% for men and women, respectively, revealing an energy deficiency for both men and women (Table 4).

On the other hand, men had a 5% protein deficiency, and women had a 20% protein surplus. Too much or too little protein was found to be associated with the risk of sarcopenia, frailty, and renal dysfunction (Table 5).

These findings suggest that, particularly for men, the intake of energy and protein is inadequate in meals provided at shelters during the middle and long-term phases of an emergency. The uncertainty surrounding meal consumption quantification persisted, given the absence of records on individual meal intake.

Additionally, in numerous instances, the entirety of the meal remained unfinished, resulting in leftover food. Regarding dietary fiber and micronutrients, the daily intake of dietary fiber and iron was estimated to be 70% and 56% for males and 82% and 65% for females (Table 6). Looking at dietary fiber and micronutrients, the daily intake of fiber and iron was estimated to be 70% and 56%, respectively, for men and 82% and 65% for women (see Table 6).

Assuming that the older adults, who accounted for 65% of the disaster-affected areas, consumed three meals a day of the 10 types of foods shown in Table 3, 120 types of combinations, and the entire amount of food rationed, the average daily energy and protein intake is shown separately for men and women. We also compared this with the recommended daily intake for people over 70 years of age. The results show that men, especially those over 70, are deficient in energy. The references of each sex and older adults’ dietary daily intake recommendations are taken from Dietary Reference Intakes for Japanese (2020) [7].

Especially for fiber, these daily intakes of 21 g for males and 18 g for females seem to be lower than the values of 25 g to 35 g proposed by the WHO. This raises concerns about the increased risk of malnutrition and frailty due to prolonged deficiencies in these essential nutrients.

## 4. Discussion

### 4.1. Two Issues of Nutrition and Health Care

Categorizing the Earthquake Aftermath into these four phases shown (Table 7), the issues related to the meals provided at the shelters were divided into two main sections: issues related to the composition of the meals served at the shelter (Issue 1); and issues related to the beneficiaries of these meals (Issue 2).

With respect to Issue 1, it was noted that meals were often high in energy, and according to the composition of meals, they were also low in fiber, far from 25 g to 35 g per day according to the WHO, which can be a challenge for older adults, especially when meals are not available for extended periods after phase 1 and 2. This situation can lead to gastrointestinal problems. Increased vigilance was required during periods of infectious disease outbreaks within the shelter. In addition, there were instances where meals were deliberately withheld to promote equity throughout the shelter.

Regarding Issue 2, which pertains to the recipients of the meals, there were occurrences where provided meals, such as those prepared communally, had to be preserved until the subsequent meal could be made stable. Additionally, there were individuals with underlying health conditions like oral fragility, underscoring the necessity for a specialized support team structure focused on dysphagia, lack of teeth, and oral hygiene. 

### 4.2. The Changes in Patients Treated and Their Causes of Diseases

The changes in the number of patients treated are shown in Figure 1. In addition, during Phase 3, there was an overlap between infectious disease outbreaks and meal arrangements in the shelters. Within the shelters, gastrointestinal symptoms were prevalent (see Figure 2), and according to the TMAT-aggregated Jspeed (Japanese minimum data set), the most common diseases throughout the period were gastrointestinal infections and food poisoning (Jspeed No. 18), accounting for 19% of cases. A high number of people had diarrhea, which worsened the state of malnutrition and the need to adapt to the diet.

In regions with a significant older adult population, gastrointestinal manifestations, especially diarrhea and vomiting, appeared to be prevalent. It is plausible that these manifestations may have been due not only to pathogens such as norovirus but also to dietary sources, such as food distributed at relief centers. Observationally, there appeared to be a remarkable proportion of individuals who experienced emetic symptoms shortly after eating. In addition, there was a lack of diagnostic tools, which made it difficult to determine whether the symptoms were related to norovirus infection or the ingestion of high-lipid meals.

### 4.3. What Kind of Meals Ought to Be Provided to High-Risk Populations, Including Older Adults, to Begin with?—The Priority of Daily Nutritional Monitoring

There is concern about refeeding syndrome when feeding is delayed for a prolonged period after a disaster, especially in older adult patients. Although there is no universal definition of refeeding syndrome, a serum P level of 2.0 mg/dL is considered “refeeding syndrome” [8]. In general, clinical experience suggests that patients are unlikely to develop refeeding syndrome due to an inability to eat for several days after a disaster and are more likely to develop refeeding syndrome, according to the A.S.P.E.N. consensus recommendation. Risk classification is based on the presence of a BMI < 18.5 kg/m^2^ and a lean body mass index (BMI > 18.5) in addition to inadequate food intake [9]. Particularly concerning immediate high energy consumption, especially glucose, which will have an impact on refeeding syndrome. Nevertheless, nutritional balance and meal distribution must be considered, especially in vulnerable populations at higher risk of gastrointestinal manifestations. This is particularly important when the timing of meal distribution, such as in soup kitchens, coincides with the occurrence of infectious diseases in shelters. The nutritional content and delivery of meals must be carefully considered. Nutritional management of vulnerable populations, such as the elderly in shelters, poses significant challenges and requires individualized approaches. Care must be extended to as many people in need as possible while recognizing the need to prioritize high-risk groups when resources are limited. For elderly individuals residing in shelters, it is advisable to initiate a low-fat dietary regimen, particularly in the context of an outbreak of infectious diseases, in order to mitigate gastrointestinal manifestations. Furthermore, it is advisable to commence with a quarter of the prescribed target to avert the occurrence of refeeding syndrome. It is difficult to deploy dietitians to evacuation shelters from the acute phase, so in an aging society with a low birth rate, daily monitoring is needed as much as possible to prioritize. In the event of an actual disaster, different teams of dietitians will be needed for different phases.

The TMAT report indicated a low incidence of acute nutritional issues in Jspeed, with just two cases documented, possibly attributable to challenges in identifying such problems during acute screening. Specifically, within the integrated aggregate report for Jspeed, only 0.159% of cases (7 out of 4381) required urgent nutritional intervention over a 12-week period [10].

### 4.4. Long-Term Shelter Stays and Its Nutritional Issues in a Super-Aging Society—The Priority of the Older Adults and the Vulnerable to Disaster

Shelter stays have often been extended in previous disasters, and during the Great East Japan Earthquake, people were required to stay in shelters for up to seven months. Consuming 1800 to 2200 kcal per day is recommended to avoid excessive or inadequate energy consumption, at least 55 g of protein, at least 0.9 mg of vitamin B1, 1.0 mg of vitamin B2, and 80 mg of vitamin C to avoid nutrient deficiencies [11]. However, it is unlikely that the shelters supported by TMAT on this occasion were adequately supplied with food to meet these guidelines. As the length of stay in the shelters increases, the need for increased support to prevent complications such as sarcopenia becomes more critical. Therefore, it might be essential to contemplate customized nutritional supplements designed for the aging population of each country and region [12]. In addition, nutrition professionals such as nutritionists and dietitians are needed to supervise and assess nutritional status and provide quality and quantity of meals, especially for older adults in acute triage.

For specific nutritional solutions to energy and protein deficiencies in older adults, we suggest that the dietitians select supplements with high energy density and protein content and select products on an individual basis.

In 2011, it was noted that a limited number of municipalities maintained operational frameworks for the procurement of specialized food supplies [13]. As indicated in the 2016 publication, the percentage of municipalities with protocols and recommendations for emergency preparedness for vulnerable persons, in addition to operational frameworks for specialized supplies, ranged from 20 to 30 percent [14,15]. Social evacuation centers are characterized by a significantly older adult population, with the primary person responsible for nutrition management being the dietician in charge. In evacuation centers with a significantly older adult population, it is believed that reduced fat content may improve appetite and reduce symptoms of postprandial vomiting. Adjustments to the diet may need to be made based on dental health and swallowing ability, possibly including foods such as porridge, soft-boiled rice, mixed meals, and nutritional supplements tailored for easy consumption by older adults because they are vulnerable to disaster due to inactivity and malnutrition, and potentially frail [16]. It should be emphasized that nutritional requirements evolve as individuals progress from the acute to the subacute and chronic stages.

In Japan, a disaster-prone country with a declining birthrate and a super-aging society, it will be necessary to consider sarcopenia and flail, and the assessment and management of nutrition in vulnerable and high-risk groups will become even more important in the future, considering that people will have to live longer in evacuation centers.

### 4.5. Strength and Limitations

The strength of this study is that TMAT, as a relief team for sudden natural disasters, was able to go to the disaster area immediately after the disaster struck. Moreover, in addition to relief activities, we were able to collect data on the health status of the victims in the disaster area and the food distributed for scientific analysis. 

This study has at least two limitations. First, we did not include nutrition experts, such as nutritionists, in the team to solve nutrition problems. This meant that we could not immediately identify and solve nutrition problems in the disaster area. Second, we were not able to stay in the disaster area long enough, which meant that we could not adequately address the deteriorating health of the elderly in the disaster area.

## 5. Conclusions

From the critical analysis of the TMAT activities of the Noto Peninsula earthquake on 1 January 2024, we evaluated the food provided in the shelters in the affected areas and found that it was not suitable for the older adults, especially regarding the energy deficiencies of 61% and 78% compared to the daily dietary requirements for men and women, from the aspect of nutritional analysis. To emphasize the urgency of implementing a framework, we would like to propose an urgent modification to the emergency food supply. In addition, it seems important to place nutritionists in the field to extract and resolve nutritional issues, especially for older adults, who comprise the majority of the victims. The limitation of this study must be warranted. The lack of nutrition experts in the team members may have led to the inability in the field to provide appropriate solutions to nutrition problems.

## Figures and Tables

**Figure 1 nutrients-16-01904-f001:**
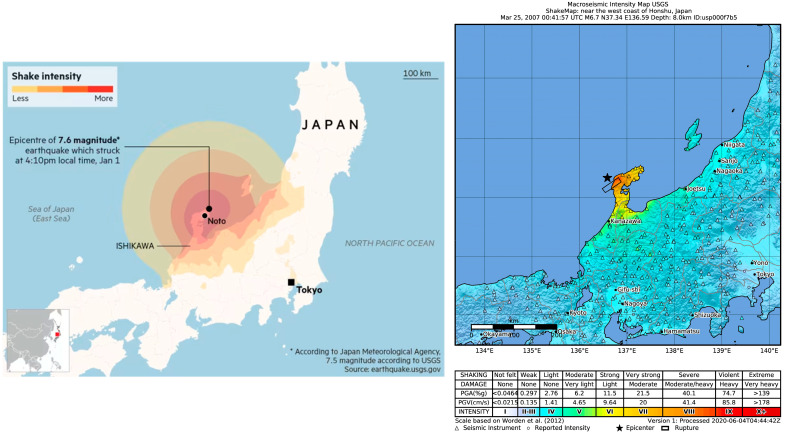
Japan’s 7.6 magnitude earthquake on the Noto Peninsula shuts the 1.2 GW coal-fired capacity [1,2].

**Figure 2 nutrients-16-01904-f002:**
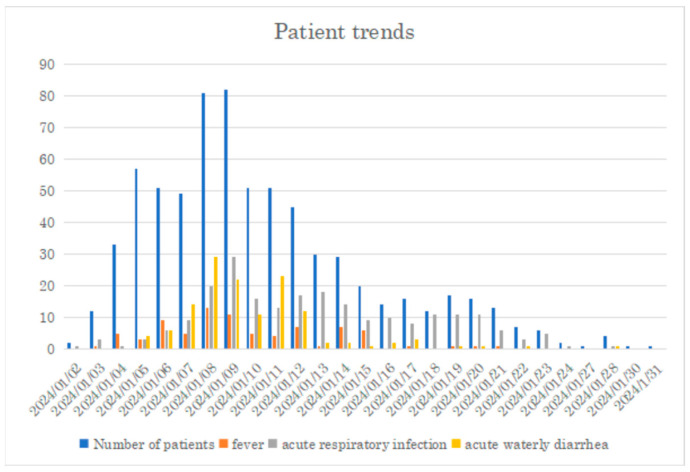
The trends of patients treated by TMAT between 2 January and 31 January 2024.

**Table 1 nutrients-16-01904-t001:** Patient profile of sex treated at TMAT.

Sex	Numbers	Percentages
male	309	44%
female	388	56%
Pregnant woman	0	0%

**Table 2 nutrients-16-01904-t002:** Patients’ profile age distribution treated at TMAT.

Age	Numbers	Percentages
0–1	0	0%
2–14	27	4%
15–64	216	31%
over 65	460	65%

**Table 3 nutrients-16-01904-t003:** Nutritional facts of food supply in the shelter.

Dish Name (Assumptions Included)	Energy (kcal)	Water	Protein	Fat	Dietary Fiber Total	Carbohydrates (g)	Sodium (mg)	Salt Equivalent (g)	Potassium (mg)	Calcium (mg)	Zinc (mg)	Vitamin B1 (mg)	Vitamin B2 (mg)	Vitamin B6 (mg)	Vitamin B12 (µg)	Vitamin C (mg)
Meat Udon Noodle (200 g noodles, including soup)	281	329	13.9	3.8	2.9	51.5	1641	4.2	222	29	1.7	0.08	0.12	0.16	0.65	4
Five-Ingredient Soba (1/3 serving)	217	224	11.4	8.1	3.4	28.3	1105	2.8	272	41	0.8	0.21	0.19	0.12	0.41	15
Champon (noodles 75 g)	150	-	9.8	2.4	-	24.1	525	1.3	207	33	-	-	-	-	-	-
Curry Donburi (230 g)	730	404	18.9	27.7	8.3	108.7	676	1.7	601	45	2.8	0.55	0.16	0.41	0.24	15
Curry Rice (200 g)	642	-	14.5	26.9	-	93.6	879	2.2	582	39	-	-	-	-	-	-
Mixed Rice with Pork Soup	438	326	21.5	7.1	5.3	76.2	1232	3.1	442	42	2	0.18	0.13	0.46	0.19	3
Chicken Rice (Assuming 200 g rice)	343	-	10.2	5	-	63.9	212	0.5	198	9	-	-	-	-	-	-
Chinese Soup	56	-	4.3	3.3	-	1.9	145	0.4	134	3	-	-	-	-	-	-
Rice (150 g), Mackerel and Vegetable Miso Stew	382	356	16.1	5.8	7.2	71.4	903	2.3	469	181	2	0.16	0.24	0.36	4.83	23
Chicken Stew with Rice (120 g)	424	-	15.6	12.3	-	66.8	256	0.7	636	45	-	-	-	-	-	-
Rice (180 g), Pork and Chinese Cabbage Stew	423	346	16	8	4.6	75.1	599	1.5	461	50	2.3	0.48	0.15	0.3	0.17	17
Rice (150 g), Canned Mackerel and Vegetable Stew	419	296	22	9.1	4.3	66.8	614	1.6	443	254	2.5	0.19	0.37	0.45	9.62	11
Rice (180 g), Chinese-style Rice Bowl	452	256	12.7	13.8	4.2	73.3	602	1.5	323	33	2	0.34	0.12	0.22	0.17	8

**Table 4 nutrients-16-01904-t004:** Comparison of energy for men and female.

	Actual Energy Supplied	Actual Energy Supplied per kg (A) *	Estimated Energy Requirement	Recommended Energy per kg (B) *	Actual/ Recommended Ratio (A/B, %)
male	1290 kcal	21.8	2100	35.4	61
female	1290 kcal	24.1	1650	30.8	78

* Average body weight for male and female are 59.3 kg and 53.6 kg, respectively.

**Table 5 nutrients-16-01904-t005:** Comparison of protein for men and female.

	Actual Protein Supplied	Animal Protein Supplied	Plant Protein Supplied	Actual Protein Supplied per kg (A) *	Estimated Protein Requirement	Recommended Protein per kg (B) *	Actual/Recommended Ratio (A/B, %)
male	47.7	29.3	18.4	0.80	60	1.12	80
SD	2.37	2.0	0.8				
female	47.7	29.3	18.4	0.89	50	0.93	95
SD	2.37	2.0	0.8				

* Average body weight for male and female are 59.3 kg and 53.6 kg, respectively.

**Table 6 nutrients-16-01904-t006:** Comparison of Other Nutrients for males and females.

	Salt (g)	Iron (mg)	Calcium (mg)	Vitamin B1 (mg)	Vitamin D (μg)	Dietary Fiber (g)
Average	6.3	3.9	235	14.7	5.5	14.7
SD	0.51	0.32	40.1	0.08	1.48	0.9
Requirement for male	7.5	7.0	750	1.0	9.0	21
% DI for male	84	56	31	1470	61	70
Requirement for female	6.5	6.0	650	0.8	9.0	18
% DI for female	97	65	36	1838	61	82

Abbreviation, DI: dietary intake, SD: standard deviation.

**Table 7 nutrients-16-01904-t007:** Disaster management cycle.

Phase	Naming of Phase	Time after Disaster Strikes	Characteristics of Food Available to Evacuees
1	Immediate response	0–24 h	Survivors may consume any available food to meet immediate energy needs.
2	Emergency Response	24–72 h	Communal cooking was prevalent, but in this instance, instant noodles became the primary food source due to regional variations.
3	Recovery and Rehabilitation	72 h to 1 month	Lack of sanitation infrastructure leads to increased digestive issues among shelter residents, coinciding with the potential spread of COVID-19. Meanwhile, cooked meals are gradually reintroduced. However, considering the limitations in distributing food to all shelter evacuees, there are cases where meals are intentionally not provided within shelters.
4	Long-term Recovery	1 month onwards	Disaster response presents distinct challenges regarding food assistance and healthcare needs like bedsores due to reduced mobility or limited access to healthcare in welfare shelters.

## Data Availability

The datasets generated and analyzed during the current study are available from the corresponding author upon reasonable request.

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
