# Peer review of "Meals in Shelters during Noto Peninsula Earthquakes Are Deficient in Energy and Protein for Older Adults Vulnerable to the Disaster: Challenges and Responses"

_nutrients, 2024, doi:10.3390/nu16121904_

Round 1

Reviewer 1 Report

Comments and Suggestions for Authors

I consider this article “Meals in shelters during the Noto Peninsula earthquakes are deficient in energy and protein for elderly people vulnerable to disaster: Challenges and responses” to be very relevant because this is a pressing issue in these populations where these disasters occur, forcing people to take shelter for very long periods outside their homes, becoming dependent on others to survive. I believe it is important to draw the attention of those responsible and propose solutions. I would therefore like to make some comments that I believe may contribute to the discussion of this article:

Item 105 ….1295 kcal of energy and 47.8 g of protein…..Protein of plant and animal origin? Wouldn’t it have been important to separate one from the other?

Item 147 …..often rich in energy…..And according to the composition of meals, they are also low in fiber, far from the 25 g to 35 g per day according to the WHO.

Item 157 …..oral hygiene….Dysphagia, lack of teeth?

Item 166 ….of cases…...High number of people with diarrhea, which will worsen the state of malnutrition and the need to adapt the diet

Item 174 …..Assistance………Need for nutritionists to supervise the entire process of preparation, distribution, storage and temperature of meals?

Item 185 ……..inadequate food intake… Particularly with regard to immediate high caloric consumption, especially glucose, which will have an impact on refeeding syndrome

Item 201 ……problems during acute triage… It is essential to screen for nutritional risk.

Item 232

….it will be necessary to consider… Malnutrition, sarcopenia….

Item 256 …especially men…..As well as women….

I would like to have had an answer to your question: What type of meals should be provided to high-risk populations, including older adults, to begin with? They should present proposals for the future not only regarding the nutritional composition of more balanced and adapted meals, but also proposals regarding timing and supervision of food hygiene and safety. As well as proposals for nutritional screening. You did not mention hydration in your study, which is essential, also taking into account the pathologies identified.

Author Response

To Reviewer 1

We would like to make responses one-by-one fashion.

The reviewer’s comments or questions are written in red ink and our responses are written in blue ink.

Reviewer 2 Report

Comments and Suggestions for Authors

1.     This research has dual values of humanitarian care and academic research. It is recommended to adjust or enhance some data in this manuscript.

2.     Except for Table 1, other tables and figures are unclear. In particular, it should not be presented in the form of screenshots and stickers. It is recommended to remake it.

3.     The sample number (N) of respondents in this study is high, and it is first-hand survey data of actual disaster victims. There should be no problems with the reliability and validity of the data. However, the reliability and validity in this study still need to be mentioned in "Materials and Methods" or "Results" and "Discussion".

4.     Can the standard deviation (SD) be presented for the dietary intake and intake of each nutrient collected from the respondents? If so, the discussion and cross-analysis will be more clear and detailed.

5.     The conclusion of this manuscript is relatively weak and should be presented with quantitative values. In addition, the limitations of this study can also be stated in the conclusion.

Author Response

To Reviewer 2

We would like to make responses one-by-one fashion.

The reviewer’s comments or questions are written in red ink and our responses are written in blue ink.
